# The Synergistic Impact of Arbuscular Mycorrhizal Fungi and Compost Tea to Enhance Bacterial Community and Improve Crop Productivity under Saline–Sodic Condition

**DOI:** 10.3390/plants13050629

**Published:** 2024-02-25

**Authors:** Fatma M. El-maghraby, Eman M. Shaker, Mohssen Elbagory, Alaa El-Dein Omara, Tamer H. Khalifa

**Affiliations:** 1Soil Microbiology Research Department, Soils, Water, and Environment Research Institute (SWERI), Agriculture Research Center (ARC), Giza 12112, Egypt; microfatma2011@gmail.com; 2Soil Improvement and Conservation Research Department, Soils, Water, and Environment Research Institute (SWERI), Agriculture Research Center (ARC), Giza 12112, Egypt; e_sh_alla@yahoo.com; 3Department of Biology, Faculty of Science and Arts, King Khalid University, Mohail 61321, Assir, Saudi Arabia; mhmohammad@kku.edu.sa

**Keywords:** mycorrhizal fungi, compost tea, diazotrophic bacteria, soil respiration rate endophytic bacteria, plant growth parameters, soil characteristics, salt-affected soil

## Abstract

Soil salinity has a negative impact on the biochemical properties of soil and on plant growth, particularly in arid and semi-arid regions. Using arbuscular mycorrhizal fungi (*Glomus versiform*) and foliar spray from compost tea as alleviating treatments, this study aimed to investigate the effects of alleviating salt stress on the growth and development of maize and wheat grown on a saline–sodic soil during the period of 2022/2023. Six treatments were used in the completely randomized factorial design experiment. The treatments included Arbuscular mycorrhizal fungus (AMF_0_, AMF_1_) and varied concentrations of compost tea (CT_0_, CT_50_, and CT_100_). AMF colonization, the bacterial community and endosphere in the rhizosphere, respiration rate, growth parameters, and the productivity were all evaluated. The application of AMF and CT, either separately or in combination, effectively mitigated the detrimental effects caused by soil salinity. The combination of AMF and CT proved to be highly efficient in improving the infection rate of AMF, the bacterial community in the rhizosphere and endosphere, growth parameters, and grain yield of maize and wheat. Therefore, it can be proposed that the inoculation of mycorrhizal fungi with compost tea in saline soils is an important strategy for enhancing salt tolerance in maize and wheat plants through improving microbial activity, the infection rate of AMF, and overall maize and wheat productivity.

## 1. Introduction

The salinization of soils has experienced rapid growth as a result of changing environmental conditions, encompassing approximately 800 million hectares of land [1]. In Egypt, salt-affected soils occupy 56% of the cultivated land under the study area (Kafr El-Sheikh Governorate) [2]. According to FAO, the excessive accumulation of soluble salts has impacted over 6% of land worldwide [3]. Plant growth in soil affected by salt is impeded by a complex interplay of factors, including ion toxicity, oxidative stress, osmotic stress, and nutrient imbalance, which individually or collectively affect the health of plants. Microbial interactions that are present within the rhizosphere are fundamental elements of a healthy soil. Consequently, all microorganisms that are associated with a plant’s endosphere, rhizosphere, and phyllosphere play a pivotal role in the growth and overall health of the plant [4,5]. To further enhance these beneficial microbial interactions, the inoculation of plants with advantageous microorganisms serves a supportive function [6]. Within this ecosystem, mycorrhizal fungi and plant growth-promoting bacteria (PGPB) emerge as two prominent contributors [7,8]. Mycorrhizal fungi possess the ability to modify the diversity and community structure of rhizosphere bacteria through alterations in soil enzyme activities and rhizosphere nutrition. Additionally, they augment plant resistance to both biotic and abiotic stresses, as well as boost the synthesis and distribution of plant hormones [9,10]. However, plants can overcome salt stress by engaging with various soil microorganisms, such as Arbuscular mycorrhizal fungi (AMF).

Arbuscular mycorrhizal fungi (AMF) have elicited growing interest as a highly efficient alternative for promoting plant growth and enhancing stress tolerance. Mycorrhizae infect the root system of the host plant, which will produce intensively interwoven hyphae [11]. Consequently, mycorrhizal plants are capable of enhancing their capacity for nutrient and water absorption [12], increasing plant resistance to soil salinity stress [13] by stimulating the photosynthetic apparatus, and enhancing the effectiveness of their antioxidant defense system. In addition, Arbuscular mycorrhizal fungi (AMF) possess the capacity to exert influence on plant-growth-promoting rhizobacteria (PGPR) in both the plant rhizosphere and hyphosphere, as highlighted by [14]. The collaboration between diazotrophs and AMF has been shown to generate a synergistic effect, enhancing each other’s growth and promoting plant growth [15]. This symbiotic relationship also serves as an efficacious and economically viable strategy for enhancing plant tolerance against salinity-induced stress, rendering it a highly valuable approach in the realm of sustainable agriculture [16]. In addition, the fertilizer foliar spraying application has become more prevalent in agricultural crop production because it is more environmentally friendly when compared to soil fertilization techniques [17]. The foliar application of compost tea (CT), which is rich in humic acids, hormones, amino acids, vitamins, minerals, and beneficial bacteria, is one of the most significant of these alternative techniques. CT has the ability to enhance the growth and yield of crops and their resistance to disease [18,19]. Furthermore, CT can be used to promote the elongation of roots and growth by producing cytokinins and gibberellic acids, these conditions can subsequently offer soil microbes a larger chance to release microbial mucilages, which support fungal hyphae and root networks in the soil, supplying a magnificent possibility for AMF to improve the infection rate. The combination of organic matter and AM fungi has been shown to play a vital role in increasing the mycorrhizal infection rate and its impact on plant growth, according to Steinberg and Rillig [20].

To address these gaps in knowledge, we formulated a hypothesis postulating that the synergistic impact of arbuscular mycorrhizal fungi (AMF) and the application of compost tea (CT) through foliar spray would lead to direct and/or indirect alterations in the bacterial community present in the rhizosphere and endosphere. Among the various bacteria inhabiting the rhizosphere, endophytes surpass others by providing plants with direct nutritional support. Consequently, this study aimed to assess the abundance and specific characteristics of endophytic diazotrophic bacteria, as these shifts and traits may play a crucial role in mitigating salt stress and promoting the growth of plants. By investigating both mycorrhizal activity and the bacterial community in the rhizosphere and endosphere, the study seeks to shed light on the effects of AMF and foliar spray of compost tea in alleviating salt stress in maize (a salt-sensitive plant) and wheat (a salt-tolerant plant), thereby influencing the overall growth and productivity of these plants.

## 2. Results

### 2.1. Biological Activities

#### 2.1.1. Mycorrhizal Colonization, Bacterial Community and Soil Respiration Rate (SRR)

The highest levels of AMF infection in maize and wheat plants were observed in the dual combination of AMF inoculated with CT_100_ (73.33–81.67%), respectively, in comparison to the remaining treatments (Figure 1). The relationships between AMF colonization and total bacterial counts, diazotrophic bacterial counts, and soil respiration rates in the rhizosphere, root, stem, and leaves of maize (Figure 2a) and wheat plants (Figure 2b), respectively, affected by AMF inoculation and the foliar spray of CT, were investigated using a correlation heatmap. The bacterial community and the rate of soil respiration were positively correlated with AMF colonization; there was a stronger association with wheat plants than with maize plants.

#### 2.1.2. Endospheric Bacterial Community

The enumeration of the endophytic bacterial population was conducted on three different tissues, namely the root, stem, and leaves for maize and wheat plants (Figure 3a,b). The total load of endophytic bacteria exhibited variation ranging from 4.16 to 5.95 log cfu g^−1^ of the maize tissue (Figure 3a), while in wheat, it ranged from 4.17 to 6.36 log cfu g^−1^ of the tissue (Figure 3b). In both maize and wheat plants, the average total load of endophytic and diazotrophic bacterium was found to be the highest in the roots, followed by the leaves and stem.

On the respective growth mediums, a total of 52 morphologically diverse isolates from maize and wheat were purified. In total, 4 isolates were from the control, whereas 48 were from AMF and CT treatments. These isolates were grouped based on Gram staining, including five Gram-positive bacterium, and 47 Gram-negative bacteria. SEM images showed that 44 isolates are rod-shaped and 8 isolates are bacilli.

### 2.2. Multivariable Analysis

The utilization of multivariate analysis techniques was employed in order to comprehend the intricate relationships among the phenotypic analyses conducted, as well as to facilitate the classification of the isolated lines. These analyses proved to be of utmost importance, as they provided invaluable assistance to plant breeders in their pursuit of understanding the genetic foundation of their isolates, particularly under conditions of diverse environmental stress.

#### 2.2.1. Principal Component Analysis (PCA) Analysis 

Loading plots of PCA graphs presented in the horizontal axis indicated the direction of association among the studied traits. For maize isolates, the loading PCA plot graph of all studied traits, as shown in Figure 4, showed that the first and two principal components described 49.6% (PCA1 = 36.3% + PCA2 = 13.3%) of the total variability, which noted that the HCN, bio-control activity, amylase activity, NaCl and siderophores traits, were found in the right side (positive) of the horizontal axis according to their positive correlations with most other traits (Figure 4a). Concerning wheat isolates, the PCA loading plot graph showed the results for all studied traits (Figure 3b). The first and second PCA accounted for 47.1% (PCA1 = 32.7% + PCA2 = 14.3%) of the total variability, which illustrious that the HCN, bio-control activity, chitinase, P, nitrogenase, and siderophore traits were found in the right side (positive) of the horizontal axis according to their positive correlations with most other traits.

#### 2.2.2. Heatmaps Cluster Analysis

In order to help the plant tolerate salinity, heatmap cluster analysis was used to visually represent some characteristics among all 52 isolates of wheat and maize, as shown in (Figure 5). Based on the 12 mean performances of traits located on the column dendrogram, all 52 isolates were divided into six clusters, with the most closed isolates being maize isolate no. 20, 18, 7, and 2, and the highest isolates being wheat isolate no. 24, 22, 15, 27, 29, 11, and 14 with high mean performances. 

We observed that the effectiveness of the isolates experienced an enhancement during the subsequent season (specifically in wheat plants). These results suggest that the application of compost tea (CT) and arbuscular mycorrhizal fungus (AM) increased the population of endophytic bacteria and produced new isolates with considerable increased inefficiency.

### 2.3. Maize and Wheat Yield and Traits

The two-way ANOVA showed that maize grain, straw, plant height, and 100-grain weight were significantly influenced by AMF, and foliar spray was influenced by compost tea (Table 1). Compared with the control treatment, individual or dual application of AMF and foliar spray by CT_100_ significantly increased the grain yield of maize by factors of 22.79%, 21.39%, and 53.41%, respectively. A similar trend was also perceived in straw yield, plant height, and 100-grain weight, which were obtained through individual or dual application of AMF and foliar spray by CT_100_. Wheat yield and traits also responded favorably to the application of AMF and compost tea (Table 1). In treatments receiving CT_100_, the GY, SY, PH, and 100-grain weight were increased by 18.58%, 16.76%, 20.84%, and 8.49%, respectively. In addition, the treatments receiving AMF increases in the same previous traits were 17.53%, 21.80%, 10.43%, and 7.80%, respectively. The most effective treatment of CT_100_ + AMF showed considerable effects on wheat yield and traits, followed by CT_50_ + AMF, AMF alone, CT_100_ alone, CT_50_ alone, and the control treatment. 

### 2.4. Nutrient Content in Maize and Wheat Plants

The application of CT and AMF and their interactions had a significant impact on the content of N, P, and K in maize and wheat plants (Table 1). After applying AMF and CT individually, maize plants had higher N, P, and K contents than the control treatment. The dual application of AMF × CT_100_ or/and CT_50_ significantly increased N, P, and K content by 49.31%, 20.38%, and 35.75%, respectively. 

Considering wheat plants, different treatments considerably improved nutrient content in wheat plants compared to the control (Table 1). With regard to different treatments, the CT_100_ foliar application treatment significantly improved N, P, and K content, which attained values of 2.44%, 0.730%, and 3.83%. High significance was observed in the application of AMF for nutrient content in wheat plants, with values of 2.42% for N, 0.747% for P, and 3.86% for K. Compared with the control treatment. N, P, and K contents in wheat plants were significantly increased by the dual application of AMF + CT_100_.

### 2.5. Soil Characteristics

#### 2.5.1. Chemical Characteristics

The two-way ANOVA showed that soil pH, organic carbon, total nitrogen, available phosphorus, potassium, and calcium were significantly influenced in both seasons by individual application of AMF treatment or a combination with CT foliar spray. Still, soil pH in the first season was not affected by these combinations (Table 2). Soil pH, organic carbon, total nitrogen, available phosphorus, and potassium were significantly increased with the AMF treatment compared with the control treatment post harvest of both crops, while the available Ca was significantly decreased. In contrast, the foliar spray of CT significantly influenced SOC, T. N, A-P, A-K, and A-Ca post harvest of both crops. 

#### 2.5.2. Biological Characteristics 

The total bacterial counts and total diazotrophic bacteria counts were insignificantly affected by the application of CT post harvest of maize or wheat. 

Inoculated AMF had total bacterial counts and diazotrophic counts ranging from 7.08 to 5.10, log CFU g^−1^ dry soil for maize, and 7.27, 5.20, and 5.13 log CFU g^−1^ dry soil for wheat. Total bacterial counts were insignificantly affected due to the combined foliar application of CT with AMF post harvest of maize or wheat, while diazotrophic counts post harvest of both crops were significantly affected by the combined treatment. AMF inoculation treatments with foliar spray by CT treatments increased diazotrophic counts post harvest of maize or wheat plants (Table 2).

## 3. Discussion

One of the key factors limiting plant growth and keeping them unproductive is soil salinization [21]. AMF incorporation in soils is one of the essential approaches to enhancing plant tolerance to salinity stressors by improving plant nutrient absorption and ion balance [22]. Moreover, AMF has been shown to protect soil enzymes and organic matter while simultaneously increasing water uptake [23,24].

Our study showed that no colonization by AMF was discovered in the roots without AMF seedlings. In AMF associations, the fungus is completely dependent on plant growth and carbohydrate nutrition production in the host plant; therefore, every factor that affects carbohydrate production and its translocation to the roots could affect the amount of mycorrhizal colonization [25].

Since salinity reduces plant growth and decreases carbohydrate concentration in host plants [26], it could reduce mycorrhizal colonization. On the other hand, the introduction of arbuscular mycorrhizal (AM) fungi directly impacts the augmentation of the mycorrhizal infection rate. Furthermore, the results of this investigation illustrate a notable enhancement in mycorrhizal infection rates when AMF is combined with compost tea (CT), as opposed to the separate application of AMF and CT treatments. This can be attributed to the synergistic effect of AMF and CT and foliar spray, which promotes root elongation and growth by stimulating the production of cytokines and gibberellic acids that enhance the colonization of AMF by expanding the root surface area and increasing the plant’s susceptibility to AMF hyphae penetration.

The presence of AMF intensively stimulates growth, increasing mycorrhizal fungal products that positively influence bacterial abundance. AMF plays a significant role in stimulating the microbial community through the production of fungal products like glomalin and glycoprotein [27], which serve as an essential energy source for bacterial communities. AMF can also perform an essential role in increasing the population of diazotrophs in the plant rhizosphere and hydrosphere [14,28]. For instance, AMF can mainly produce hyphae to offer highly efficient corridors for available nutrients, thus favoring nutrient communications with diazotrophs [29] or promoting the growth of heterotrophic nondiazotrophs to modify the community of diazotrophs [30,31]. The diversity of bacterial communities in the rhizosphere can be attributed to the measured levels of CO_2_ and properties of the plant. The rise in soil respiration is an indication of an increased presence of microbial communities in the soil following the introduction of AMF and CT. These findings align with the observations made by [32], who noted that the existence of AM and compost wastes leads to higher levels of CO_2_ emission.

For instance, previous research by Rodríguez-Caballero et al. [33] demonstrated that the introduction of AMF resulted in a shift in the root bacterial community, which in turn affected shoot phosphorus uptake. Conversely, it has also been demonstrated that AM fungi possess the capacity to either stimulate or inhibit the growth of specific microbes [34] through the provision of carbon compounds derived from host assimilates to mycorrhizospheric bacteria via mycelia, engaging in nutrient competition with bacteria, and releasing inhibitory or stimulatory compounds [35,36]. In line with these findings, alterations in the root microbiome have been reported following AM fungal inoculation. For instance, Solís-Domínguez et al. [37] utilized denaturing gradient gel electrophoresis to reveal that AM fungal inoculation induced changes in the bacterial and fungal communities associated with mesquite plants grown in a greenhouse environment containing desert mine tailings.

In addition to the dynamic role of AMF in increasing levels of endophytic bacteria, CT can play a critical role in stimulating microbial community. Pariona-Llanos et al. [38] showed that organic fertilization enhances the number of diazotrophic endophytic bacteria from roots, stems, and leaves when compared with conventional fertilization.

AMF is closely associated with large and diverse bacterial communities, which may colonize spores, sporocarps, and extra radical hyphae, originating a complex and metabolically active environment called the mycorrhizosphere [39]. Microbiota possess various plant growth-promoting properties (plant hormone, antibiotic, and siderophore production, N_2_ fixation, P solubilization), as well as mycorrhizal establishment facilitation (spore germination and mycelial growth promotion, mycorrhizal establishment facilitation). These agreements with our results, from PCA analysis revealed that AMF inoculation has the potential to induce alterations in the endogenous bacterial composition. Moreover, AMF symbiosis is accompanied by various modifications in gene expression within the roots and shoots of mycorrhizal plants, which are associated with numerous physiological functions of the plants [40]. Apart from these direct effects of AM fungal inoculation, introduced AM fungi may also exert an indirect influence on the root microbiome by altering the patterns of root exudation, resulting from improved plant health status [41]. 

In this study, the most significant enhancements in nutrient uptake; specifically, N, P, and K were observed when AMF and compost were applied together. The benefits were comparatively less pronounced with AMF inoculation alone, and even more limited with compost treatments alone. These improvements resulting from the spreading of AMF hyphae, beyond the root zone, facilitate the absobation of P and other immobile nutrients to plants and thereby promote biomass production [42].

AMF, native microflora, and compost enhanced the development of yield, traits, and the majority of mineral content in maize and wheat plants due to the combination of a foliar application of compost tea and AMFs [43], who observed that AMF inoculation increased distribution of photosynthate in structures, and mycorrhizal roots could release more root exudates and improved nutrition. Similarly, the inoculation of AMF greatly increased wheat grain yield, due to enhanced rhizosphere acidification and nutrient acquisition [44,45]. The combined application of AMF and CT significantly increased the growth, yield, and nutrient uptake of wheat [46].

The improvement in soil biochemical properties is attributed to a combination of various mechanisms resulting from the synergistic interactions between AMF and diazotrophic bacteria [7,47], while CT aids in promoting the activities of both populations [48,49]. There was a decrease in soil nutrient content due to the high nutrient uptake promoted by AMF, particularly P [50,51]. Sun et al. [52] showed that soil N, available P, and soil organic C content were influenced by the application of AMFs. 

After harvesting, we noted that adding AMF increased the soil pH, possibly because AMF directly affects the quantity and quality of root exudates. These exudates improve the soil bacterial community and increase bacterial metabolites, which increase the soil pH. We also noticed that soil P increased even though the soil pH was high, resulting from the production of glycoprotein glomalin by AMF, which binds the soil particles into aggregates leaving P in the soil even after mycorrhizal death [53]. 

## 4. Materials and Methods

### 4.1. Materials

#### 4.1.1. Soil

From the selected field experiment (Sakha Agricultural Research Station, Kafr El-Sheikh Governorate, Egypt, located at 31 05′08″ N, 30°56′23″ E with an elevation of 4.28 m above mean sea level), a soil sample (0–30 cm) was collected for analysis before maize was sown. The initial soil properties (Table 3) indicated that the soil was a saline–sodic heavy clay soil with low organic carbon and nutrient content.

#### 4.1.2. Arbuscular Mycorrhizal Fungi Inoculum

The propagule arbuscular mycorrhiza (*Glomus versiform*, SWERI 112) was kindly provided by the Soils, Water and Environment Research Institute (SWERI) at the Agricultural Research Center (ARC) in Giza, Egypt. To produce a large quantity of AM spores, a mixture of sterilized sand and soil (in a ratio of 1:1) was placed in pots, with each pot containing 6 kg of the mixture. Subsequently, onion bulbs were cultivated in these pots, which contained mycorrhizae inoculum. The plants were then irrigated according to their water requirements, which were determined through visual observations of the potted plants and the soil. After three weeks, the onion plants were harvested. The AMF propagules were gathered from the soil using the wet sieving and decanting method, and they were counted according to the protocol outlined by [54]. The inoculum material comprises f 109 spores per gram, in addition to fragments of colonization roots.

Maize and wheat seeds were surface-sterilized with a solution of 3% NaClO for approximately 2 min. Subsequently, they were washed five times with sterile distilled water containing a few drops of H_2_O_2_. The sterilized seeds of maize and wheat were initially combined with a solution of gum arabic at a concentration of 1%. Subsequently, the mycorrhiza inoculum was introduced which contained 10^9^ spores/dry soils. The mixture was then left undisturbed for 30 min to facilitate partial drying and the fixation of the inoculum onto the surface of the seeds.

Finally, two maize seeds (cv. Hybrid 368) were sown in each hole, with a spacing of 20 cm between them in the summer season of 2022. Conversely, wheat seeds (cv. Sakha 94) were sown in broadcast in the winter season of 2022/2023.

#### 4.1.3. Compost Tea

To produce compost tea, compost was gratefully received from the Agricultural Microbiology Department, Agricultural Research Center, Sakha, Kafr El-Sheikh, Egypt. It was prepared by brewing compost and water at a ratio of 1:10 *w*/*v* (compost:water). with continuous aeration for 48 h. To eliminate the presence of chlorine, tap water was added to the brewing tank approximately 48 h before its use to allow for volatilization, as mentioned by [55]. Molasses (10 mL/L) was added initially during the brewing process as a carbon supplement for the enrichment of beneficial microbial growth in the compost tea. Following this, the compost tea was filtrated. The CT characteristics during the two seasons were as follows: pH, (7.3 and 7.5); EC (2.47 and 2.77 dS m^−1^); total N (5120 and 5280 ppm); available P (3350 and 3719 ppm); available K (4346 and 4514 ppm); total count of bacteria (7.9 and 7.5 log cfu mL^−1^); total count of actinomycetes (4.65 and 4.88 log cfu mL^−1^); total count of fungi (4.11 and 4.00 log cfu mL^−1^); diazotrophic counts on JNfb (6.4 and 6.14 log cfu mL^−1^) and on LGI (6.14 and 5.8 log cfu mL^−1^), respectively.

### 4.2. Experimental Setup

In completely randomized factorial design, 2 factors were studied: (1) Mycorrhizal fungi inoculation of AMF_0_, and AMF_1_: none, and inoculation, respectively. (2) Compost tea foliar spray of CT_0_, CT_50_ and CT_100_: none, 50%, and 100% from the recommended dose (100 L ha^−1^) [56] to examine the ameliorating effects on the bacterial community in rhizosphere and endophytic tissues. These effects were analyzed with the aim of assessing microbial activity, growth parameters, and productivity of maize and wheat plants under salt stress. Each treatment had a plot area of 10 m^2^ (4 m × 2.5 m) with 3 replications. The foliar treatment was sprayed at 30 and 50 days of growth stage using a hand atomizer. 

### 4.3. Measurements and Analyses

#### 4.3.1. Biological Activities

##### Assessment of Arbuscular Mycorrhizae Fungi Infection

To evaluate the colonization of AMF, fine roots were collected from the lateral root system at the flowering stage. The technique developed by [57] was employed to stain root samples and subject them to a digital computerized microscope at 40–10× magnification examination in order to determine the colonization percentage. In addition, the following equation was employed by [58] to estimate the percentage of AMF infection.
(1)$$\%\;Colonization=\frac{Total\;number\;of\;AMF\;positive\;segments}{The\;total\;number\;of\;segments\;studied}\times 100$$

##### Bacterial Community in Rhizosphere Soil and Soil Respiration Rate (SRR)

Total bacteria and non-symbiotic diazotrophic bacteria were examined in the rhizosphere of maize and wheat plants under each treatment at the flowering and harvest stage. This was accomplished using a soil dilution plate technique as described by [59]. The total count of bacteria was determined using soil extract agar media, as described by [60]. Conversely, Diazotrophic bacterial counts were assessed using LGI (sucrose as C-source) and JNfb (malic acid as C-source) agar medium, as recommended by [61]. We recorded the total number of diazotrophic bacteria obtained from the sum counts of JNFb and LGI. Soil respiration rate (CO_2_ (mg) 10 g^−1^ 72 h^−1^) was measured by [62] to quantify microbial activity. The foundation of this technique is the detection of CO_2_ generated during soil microbial activity. 

##### Counts and Isolation of Endophytic Bacteria

The dilution plate spread method counted the endophytic bacteria on the roots, stems, and leaves of plants at the flowering stage. They were surface-sterilized (disinfection solution comprising 90:5:5 of sterile deionized water, 96% ethanol, and sodium hypochlorite solution (5% active chlorine)) for 10 min, followed by five rinsings in sterile deionized water under a laminar flow cabinet. After surface sterilization, samples (1 g) were cut up, fully ground in a sterile mortar with a small amount of sterile quartz sand, and then diluted with sterile phosphate-buffered saline (pH 7.4), into suspension 10^−2^, 10^−3^, and 10^−4^. One hundred microliters were spread on the Trypticase Soy Agar (TSA) agar medium plate, and inoculated plates were incubated at 30 °C for 24–72 h for the viable counts of bacteria. In total, 100 μL aliquots from dilutions were transferred to N-free LGI and JNfb agar medium for diazotrophic counts. The bacterial isolates were selected based on morphological and biochemical parameters. Purified bacterial morpho types were stored in respective media slants at 4 °C for the working culture and as a 25% glycerol stock at 20 °C for future use.

##### Phenotypic Characterization of Isolates

The colony morphology and gram reaction of isolates were examined. The catalase test of isolates was conducted as described by [63]. Starch hydrolysis and proteolysis were assessed according to [64,65]. Lipid hydrolysis and cellulose degradation were studied as described by [66]. Collmer et al. [67] used the method for pectolysis analysis, while chitin hydrolysis was performed as described by [68]. 

##### Evaluation of Plant Growth-Promoting (PGP) Traits

The potential PGP characteristics were assessed in a manner similar to [69], with some modifications. In a brief explanation, phosphate solubilization was determined by utilizing Pikovskaya agar plates that contained tricalcium phosphate (Ca_3_ (PO_4_)_2_) as the insoluble source of phosphate. These plates were then incubated in the absence of light at a temperature of 27 ± 1 °C for the duration of 5 days. The appearance of a clear halo surrounding the bacteria indicated a positive capability for phosphate solubilization. To evaluate the production of hydrogen cyanide (HCN), nutrient broth (5 mL) was inoculated into 30 mL glass tubes that contained glycine (0.44%) along with a strip of sterilized filter paper saturated with a solution of picric acid (0.5%) and sodium carbonate (2%). Following an incubation period of 7–15 days at 30 °C, a change in the color of the filter paper, from yellow to light brown or reddish-brown, was seen as a positive indicator of HCN production [70]. The estimation of siderophore production was carried out using chrome azurol S (CAS) agar, as described by [71]. Incubation of the plates took place in the absence of light at 27 ± 1 °C for duration of 7 days. The observation of orange halos surrounding the colonies on the blue agar indicated the excretion of siderophores. In order to assess the indole-3-acetic acid production (IAA), the bacteria were cultivated in TSA that was supplemented with 100 μg/mL of L-tryptophan. The culture was then incubated at 27 ± 1 °C for duration of 2 days, with agitation at 150 rpm. Subsequently, a mixture of 2 mL of cell suspension and 4 mL of Salkowski’s reagent was incubated at room temperature in darkness for a period of 30 min, then measured at a wavelength of 530 nm as described by [72]. TSA+ 100 μg/mL L- tryptophan was used as control.

##### Biocontrol Activities against Potential Plant Pathogens

The biocontrol efficacy of the endophytic bacteria that were isolated was evaluated against Rhizoctonia solani through the use of a dual inoculation technique, as described by [73]. The plaques that were inoculated were incubated at a temperature of 30 °C for a period of 5–7 days, and, subsequently, the zone of inhibition of the bacteria against the fungal pathogens was determined. A positive inhibition rate was recorded when there was a distinct deceleration in the fungal mycelium growth compared with the control treatment where no bacterial inoculation was applied.

#### 4.3.2. Plant Analysis

After the maturity stage of both plant types, ten plants were collected randomly from each treatment to estimate plant height (cm), 100-grain weight (g), straw and grain yield (Mg ha^−1^). In addition, the NPK contents in both plants (%) were determined as described by [74].

#### 4.3.3. Soil Analysis

Soil samples were subjected to a series of procedures including air-drying, gentle crushing, and sieving through a 2 mm sieve. The soil characteristics were analyzed using established methods as outlined by [74,75]. 

### 4.4. Statistical Analysis

Through the use of Minitab software (version 21.4.1), the data analysis and Principal Component Analysis (PCA) were produced. A two-way ANOVA along with Tukey’s multiple tests were applied to examine the significant differences among different treatments of main affects and their interaction at *p* < 0.05. Heat maps and multivariate data were generated using a correlation package in R-software (R for Windows 4.3.1).

## 5. Conclusions

Individual application of AMFs and a foliar application of compost tea had positive effects on maize and wheat yield, traits, and the majority of mineral content in plants. In addition, the combination of a foliar application of compost tea and AMFs had synergistic effects on maize and wheat yield, traits, nutrient uptake, and biochemical soil properties that were performed in the present study. It was concluded that the inoculation of mycorrhizal fungi with a foliar application of compost tea at a rate of 100% in saline soils is an important strategy for enhancing the salt tolerance of maize and wheat plants through enhancing microbial activities and grain yield for maize and wheat.

## Figures and Tables

**Figure 1 plants-13-00629-f001:**
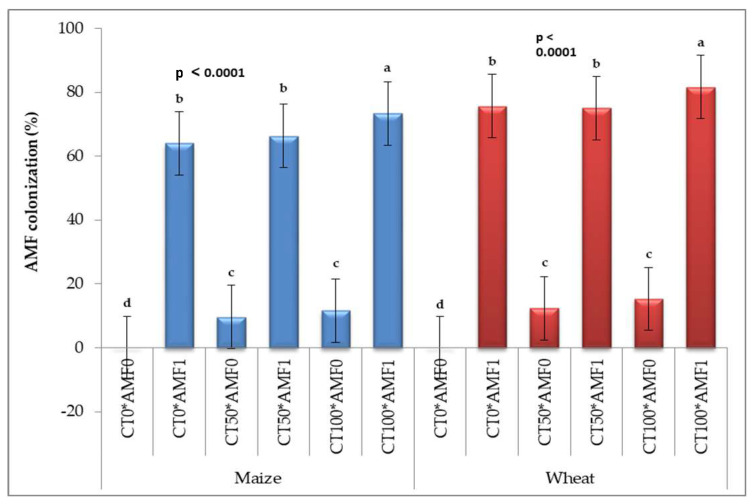
Mycorrhizal colonization in maize and wheat plants. * CT0: without compost tea; CT50: 50% compost tea; CT100: 100% compost tea; AMF_0_: without mycorrhizal fungus inoculation; AMF_1_: mycorrhizal fungus inoculation treatments, ^a–d^: Duncun letters.

**Figure 2 plants-13-00629-f002:**
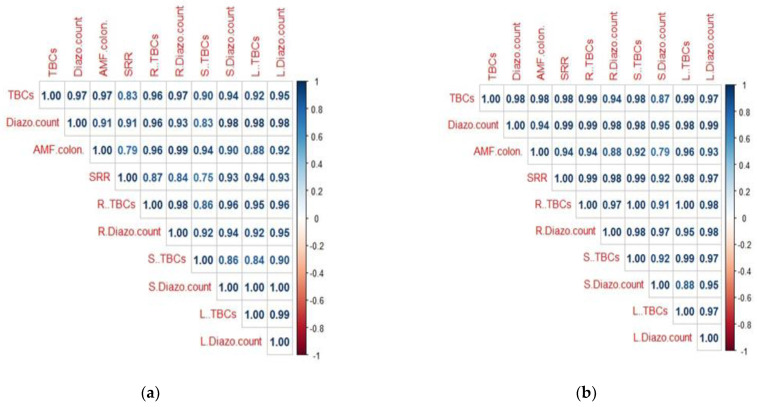
Correlation heatmap explored the relationships between Mycorrhizal colonization, bacterial community and soil respiration rate in maize plants (**a**) and wheat plants (**b**). AMF.colon: Mycorrhizal colonization, TBCs: total bacterial counts, Diazo.count: diazotrophic bacterial count, SRR: soil respiration rate in rhizosphere; R..TBCs, R. Diazo.count in root of plants; S..TBCs, S.Diazo.count in stem of plants; L..TBCs, L.Diazo.count in leaves of plants.

**Figure 3 plants-13-00629-f003:**
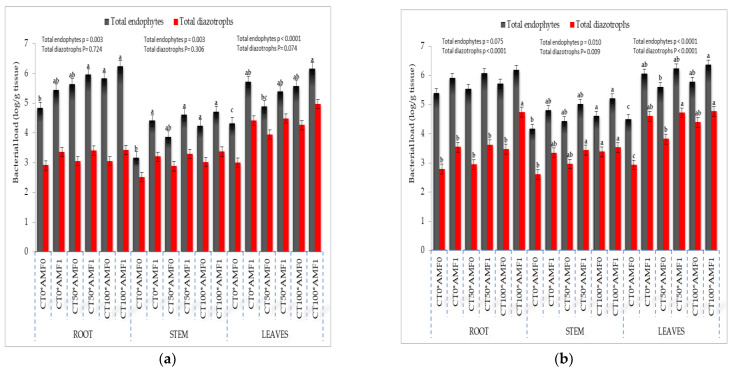
Effects of dual application of AMF and foliar spray by CT on endophytic bacteria community in maize (**a**) and wheat (**b**). * CT0: without compost tea; CT50: 50% compost tea; CT100: 100% compost tea; AMF_0_: without mycorrhizal fungus inoculation; AMF_1_: mycorrhizal fungus inoculation treatments, ^a–c^: Duncun letters.

**Figure 4 plants-13-00629-f004:**
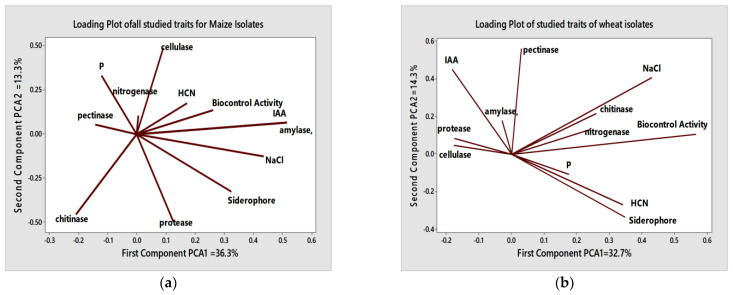
Loading plot graph, showing the first two principal components (PCA) of the correlation matrix among the studied traits for maize (**a**) and wheat isolates (**b**).

**Figure 5 plants-13-00629-f005:**
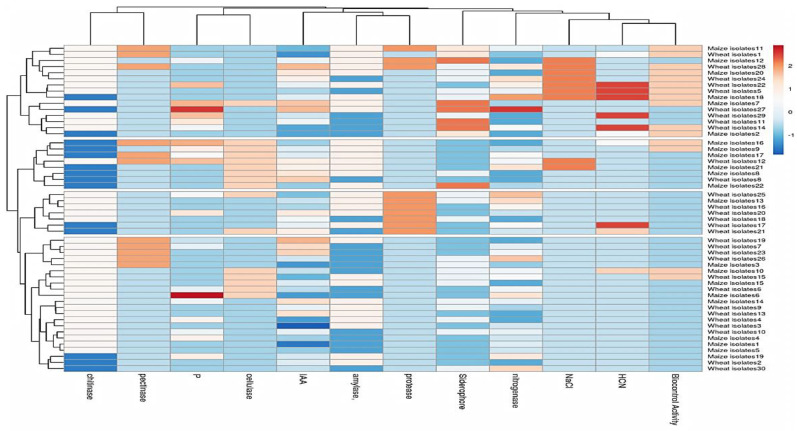
Hierarchical clustering heat map visualization of 30 wheat and 22 maize isolates using 12 traits; red cells indicate high values and blue cells indicate low values. High to low values are scaled according to the key above.

**Table 1 plants-13-00629-t001:** Two-way ANOVA used to show main and interaction treatment effects on the variable measured in maize and wheat plants.

Treatments	Maize (Season 2022)							Wheat (Season 2022/2023)						
	G.Y	S.Y	PH	100 G.W	N%	P%	K%	G.Y	S.Y	PH	100 G.W	N%	P%	K%
**Compost tea foliar spray (A)**														
CT_0_	9.23 ^c^	10.34 ^b^	255.33 ^c^	38.36 ^c^	3.17 ^c^	0.881 ^c^	2.55 ^c^	6.72 ^b^	8.81 ^c^	75.17 ^c^	4.12 ^c^	2.11 ^c^	0.583 ^c^	3.09 ^c^
CT_50_	10.50 ^b^	11.01 ^ab^	265.17 ^b^	40.63 ^b^	3.70 ^b^	0.927 ^b^	2.74 ^b^	7.70 ^a^	9.51 ^b^	86.17 ^b^	4.29 ^b^	2.21 ^b^	0.663 ^b^	3.58 ^b^
CT_100_	11.21 ^a^	11.26 ^a^	274.83 ^a^	41.88 ^c^	3.97 ^a^	0.955 ^a^	2.90 ^a^	7.97 ^a^	10.28 ^a^	90.83 ^a^	4.47 ^a^	2.44 ^a^	0.730 ^a^	3.83 ^a^
F-Value	102.9	3.72	23.77	33.95	64.18	27.35	32.46	33.85	41.69	116.47	85.79	296.61	67.68	131.4
*p*-Value	<0.0001	0.055	<0.0001	<0.0001	<0.0001	<0.0001	<0.0001	<0.0001	<0.0001	<0.0001	<0.0001	<0.0001	<0.0001	<0.0001
**Arbuscular Mycorrhizal Fungi (B)**														
AMF_0_	9.26 ^b^	10.34 ^b^	255.78 ^b^	38.24 ^b^	3.17 ^b^	0.872 ^b^	2.51 ^b^	6.86 ^b^	8.59 ^b^	79.89 ^b^	4.13 ^b^	2.08 ^b^	0.570 ^b^	3.15 ^b^
AMF_1_	11.37 ^a^	11.40 ^a^	274.44 ^a^	42.33 ^a^	4.05 ^a^	0.970 ^a^	2.95 ^a^	8.07 ^a^	10.47 ^a^	88.22 ^a^	4.46 ^a^	2.42 ^a^	0.747 ^a^	3.86 ^a^
F-Value	342.9	9.45	65.33	133.59	225.3	143.36	164.1	85.37	201.5	93.75	218.39	887.04	290.59	347.3
*p*-Value	<0.0001	0.01	<0.0001	<0.0001	<0.0001	<0.0001	<0.0001	<0.0001	<0.0001	<0.0001	<0.0001	<0.0001	<0.0001	<0.0001
**Interaction (A × B)**														
CT_0_ *AMF_0_	7.68 ^d^	9.88 ^b^	242.67 ^c^	36.16 ^d^	3.11 ^c^	0.839 ^c^	2.28 ^e^	6.19 ^c^	7.59 ^e^	72.67 ^e^	4.02 ^d^	1.88 ^d^	0.485 ^e^	2.53 ^e^
CT_0_ *AMF_1_	10.79 ^b^	10.79 ^b^	268.00 ^ab^	40.56 ^bc^	3.23 ^c^	0.923 ^b^	2.83 ^bc^	7.26 ^b^	10.02 ^bc^	77.67 ^de^	4.22 ^c^	2.33 ^b^	0.680 ^c^	3.65 ^bc^
CT_50_ *AMF_0_	9.46 ^c^	10.41 ^b^	256.00 ^bc^	38.66 ^c^	3.12 ^c^	0.876 ^bc^	2.54 ^d^	7.02 ^b^	8.43 ^d^	81.67 ^cd^	4.16 ^c^	2.04 ^c^	0.580 ^d^	3.32 ^d^
CT_50_ *AMF_1_	11.54 ^a^	11.61 ^ab^	274.33 ^a^	42.59 ^ab^	4.29 ^b^	0.978 ^a^	2.93 a	8.37 ^a^	10.58 ^ab^	90.67 ^b^	4.42 ^b^	2.39 ^b^	0.745 ^b^	3.85 ^b^
CT_100_ *AMF_0_	10.64 ^b^	10.72 ^b^	268.67 ^ab^	39.91 ^c^	3.30 ^c^	0.900 ^b^	2.70 ^cd^	7.38 ^b^	9.76 ^c^	85.33 ^c^	4.22 ^c^	2.33 ^b^	0.645 ^c^	3.59 ^c^
CT_100_ *AMF_1_	11.78 ^a^	11.79 ^a^	281.00 ^a^	43.85 ^a^	4.64 ^a^	1.01 ^a^	3.10 ^a^	8.57 ^a^	10.81 ^a^	96.33 ^a^	4.72 ^a^	2.54 ^a^	0.815 ^a^	4.08 ^a^
F-Value	119.68	4.49	23.36	40.38	42.19	39.96	46.64	30.76	61.04	67.02	85.13	311.22	85.51	133.25
*p*-Value	<0.0001	0.015	<0.0001	<0.0001	<0.0001	<0.0001	<0.0001	<0.0001	<0.0001	<0.0001	<0.0001	<0.0001	<0.0001	<0.0001

* CT0: without compost tea; CT50: 50% compost tea; CT100: 100% compost tea; AMF_0_: without mycorrhizal fungus inoculation; AMF_1_: mycorrhizal fungus inoculation treatments. GY: Grain yield (Mg ha^−1^), SY: Straw yield (Mg ha^−1^), PH: plant height (cm), and 100 G.W: 100 grain weight (g). Tukey’s test was used to group information at a *p* < 0.05. Means that do not share a letter have a significant difference.

**Table 2 plants-13-00629-t002:** The two-way ANOVA of main and interactions treatment effects on the variable measured in soil after maize and wheat harvest.

Variable Measured		Maize 2022									Wheat 2022/2023					
	pH	S.O.C	T. N	A-P	A-K	A-Ca	TBC	DZ	pH	S.O.C	T. N	A-P	A-K	A-Ca	TBC	DZ
**Compost tea foliar spray (A)**																
CT0	8.41	4.97	0.677 ^a^	7.48 ^a^	267.67 ^a^	15.49 ^a^	6.64	4.06 ^c^	8.39	4.87 ^a^	0.663 ^a^	7.51 ^a^	267.67	15.29 ^a^	6.72	4.36 ^b^
CT50	8.42	4.91	0.669 ^a^	7.38 ^ab^	261.00 ^ab^	15.34 ^ab^	6.66	4.57 ^b^	8.4	4.85 ^a^	0.660 ^a^	7.38 ^b^	263.83	15.05 ^ab^	6.85	4.90 ^a^
CT100	8.4	4.92	0.671 ^a^	7.31 ^b^	256.83 ^b^	15.20 ^b^	6.86	4.89 ^a^	8.36	4.84 ^a^	0.659 ^a^	7.35 ^b^	262	14.95 ^b^	7.04	4.97 ^a^
F-Value	0.6	1.05	1.05	10.08	5.91	4.03	0.19	18.13	1.07	0.4	0.4	14.71	0.94	6.09	0.33	12.38
*p*-Value	0.566	0.381	0.381	0.003	0.016	0.046	0.831	0.0002	0.372	0.679	0.679	0.001	0.417	0.015	0.726	0.0012
**Arbuscular Mycorrhizal Fungi (B)**																
F0	8.38 ^b^	4.80 ^b^	0.653 ^b^	7.04 ^b^	256.22 ^b^	15.51 ^a^	6.35b	3.87 ^a^	8.34 ^b^	4.76 ^b^	0.648b	6.99 ^b^	250.22 ^b^	15.25 ^a^	6.47 ^b^	4.32 ^b^
F1	8.44 ^a^	5.07 ^a^	0.691 ^a^	7.75 ^a^	266.78 ^a^	15.17b	7.08 ^a^	5.14b	8.42 ^a^	4.94 ^a^	0.673 ^a^	7.84 ^a^	278.78 ^a^	14.94 ^b^	7.27 ^a^	5.17 ^a^
F-Value	12.76	71.84	71.84	462.81	20.28	16.92	4.99	123.35	27.13	57.68	57.68	1082.04	68.94	14.44	6.13	61.01
*p*-Value	0.004	<0.0001	<0.0001	<0.0001	0.004	0.001	0.045	<0.0001	<0.0001	<0.0001	<0.0001	<0.0001	<0.0001	0.003	0.029	<0.0001
**Interaction (A × B)**																
CT0 *AMF0	8.38	4.84 ^b^	0.660 ^b^	7.11 ^c^	261.33 ^abc^	15.70 ^a^	6.18	3.18 ^d^	8.36 ^ab^	4.79 ^b^	0.653 ^b^	7.05 ^c^	257.33 ^b^	15.50 ^a^	6.25	3.68 ^d^
CT0 *AMF1	8.43	5.09 ^a^	0.693 ^a^	7.86 ^a^	272.00 ^a^	15.27 ^ab^	7.09	4.94 ^a^	8.42 ^a^	4.94 ^a^	0.673 ^a^	7.97 ^a^	278.00 ^a^	15.09 ^ab^	7.19	5.05 ^ab^
CT50 *AMF0	8.4	4.77 ^b^	0.650 ^a^	7.05 ^c^	255.33 ^bc^	15.46 ^ab^	6.32	3.97 ^c^	8.35 ^ab^	4.75 ^b^	0.647 ^b^	7.00 ^c^	248.00 ^b^	15.20 ^ab^	6.42	4.58 ^c^
CT50 *AMF1	8.45	5.05 ^a^	0.688 ^a^	7.72 ^ab^	266.67 ^ab^	15.23 ^ab^	6.99	5.16 ^a^	8.44 ^a^	4.95 ^a^	0.674 ^a^	7.77 ^b^	279.67 ^a^	14.90 ^b^	7.27	5.22 ^a^
CT100 *AMF0	8.37	4.78 ^b^	0.651 ^b^	6.96 ^c^	252.00c	15.38 ^ab^	6.56	4.47 ^b^	8.31 ^b^	4.74 ^b^	0.646 ^b^	6.93c	245.33 ^b^	15.07 ^ab^	6.73	4.70 ^bc^
CT100 *AMF1	8.43	5.08 ^a^	0.692 ^a^	7.66 ^b^	261.67 ^abc^	15.01 ^b^	7.15	5.31 ^a^	8.42 ^a^	4.94 ^a^	0.673 ^a^	7.77 ^b^	278.67 ^a^	14.83 ^b^	7.35	5.24 ^a^
F-Value	0.07	0.25	0.25	0.59	0.04	0.46	0.09	5.52	0.66	0.58	0.58	3.26	1.34	0.35	0.09	5.75
*p*-Value	0.936	0.78	0.78	0.569	0.958	0.641	0.916	0.0199	0.536	0.576	0.576	0.074	0.3	0.712	0.918	0.0178

* CT0: without compost tea; CT50: 50% compost tea; CT100: 100% compost tea; AMF_0_: without mycorrhizal fungus inoculation; AMF_1_: mycorrhizal fungus inoculation treatments. S.O.C: organic carbon (g kg^−1^), T.N: total nitrogen (g kg^−1^), A-P: available phosphorus (mg kg^−1^), A-K: available potassium (mg kg^−1^), A-Ca: available calcium (mg kg^−1^), TBC: total bacterial counts (log CFU g^−1^ dry soil) and DZ diazotrophic counts (log CFU g^−1^ dry soil). Tukey’s test used to group information at a *p* < 0.05. Means that do not share a letter have a significant difference.

**Table 3 plants-13-00629-t003:** Initial soil properties before maize sowing (2022).

Soil Properties			Available Macro Nutrients (mg kg^−1^)		
pH	ECs (ds m^−1^)	ESP (%)	P	K	Ca
8.37	7.94	19.44	7.12	266	16.1
		Particle Size Distribution (g kg^−1^)			
S.O.C (g kg^−1^)	Total N (g kg^−1^)	Clay	Silt	Sand	Texture grade
6.85	0.59	523.9	296.8	179.30	Clayey
Biological properties (log cfu g^−1^ dry soil)					
Total microbial count			Diazotrophic counts		
Bacteria	Actinomycetes	Fungi	JNfb	LGI	
7.67	4.20	4.35	3.72	4.3	

## Data Availability

The data that support the findings of this study are contained within the article and available from the corresponding author upon reasonable request.

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
