# Peer review of "The Synergistic Impact of Arbuscular Mycorrhizal Fungi and Compost Tea to Enhance Bacterial Community and Improve Crop Productivity under Saline–Sodic Condition"

_plants, 2024, doi:10.3390/plants13050629_

Round 1
Reviewer 1 Report
Comments and Suggestions for Authors
The reviewed research is of great importance, not only theoretically but above all, practically. Therefore, it is worth emphasizing this in the introduction, in the hypotheses.
Line 36: instead of 'envi-ronmental conditions' should be 'environmental conditions'
Lines 74-80: please reword your hypothesis as it is not very clear in this form. It would also be worth including in the hypotheses the possibility of synergistic effects of AMF and foliar application of compost tea
Line 97: please explain the abbreviations used in the caption under Figure 1; the same in the case of Figure 3 (line 118)
Line 169, Table 1: please explain the abbreviations in the first column; the same in the Table 2 (line 222);
Lines 354-362: it would be worth providing the total number of samples and a graphical scheme of the experiment

Reviewer 2 Report
Comments and Suggestions for Authors
This paper conducted AMF and CT treatments on two crops, attempting to study the effects of these treatments on the resistance of crops to salt stress. The authors carried out measurements and analysis from many aspects; however, the provided data is too rough and of low quality, and lacks essential details. It is insufficient to support the results claimed by the authors.
Figures are relatively independent parts; therefore, it is necessary to provide details to help readers understand. However, Figure 1 is missing a figure legend. Similarly, many other images lack necessary captions, too.
Figure 1 and Figure 3 lack significance analysis, so they cannot explain any issues. Figure 5 is unclear; is it showing isolated microorganisms?
This paper does not explain how the spores of AMF are produced.
P11, “Finally, two maize seeds (cv. Hybrid 368) sown in each hole”. As for the maize seeds treated with spores of AMF, during sowing, are they sown alone, or are they sown along with soil containing Spores of AMF?
The format of Table 3 is not standardized.
P11, “Compost tea was prepared by brewing compost and water at a ratio of 1:10 w/v (compost: water) with continuous aeration for 48hour,” The authors did not mention how to prepare the compost.
Reviewer 3 Report
Comments and Suggestions for Authors
Dear Authors,
my main comment and suggestion is the lack of description in the introduction of examples of specific species and specific examples from the literature of the described issue. I think this is missing in the introduction. As for the results and summary, I understand the idea of the work and it is coherent. It would be worth going into greater detail, but since the work at this stage is coherent and substantively correct, I do not believe that this should limit the publication of the manuscript.
Reviewer 4 Report
Comments and Suggestions for Authors
The authors have provided an account of a study of the impact of VA inoculation and compost tea on maize and wheat in saline soils.
While the results of the individual parts of the study all have some scientific merits, they are presented in a way that makes it hard to find a common thread that goes beyond the presentation of unconnected studies.
Many parts of the manuscript are confusing to read. There is no clear explanation of the heatmap in Figure 5 and how it contributes to the overall context of the study.It appears that the authors try to connect several different studies without a clear overall context.
The discussion needs to be significantly expanded and improved, with clear references to the authors' findings.
I would suggest to completely rewrite the manuscript. Also, provide more context to salinity in an expanded discussion. E.g. how saline are these soils in the context of other regions? How do yields compare to non-saline soils with inoculation?
Results, discussion and methods must be clearly connected so that the reader can see the overall "narrative" of the manuscript.
Other comments:
The scientific name is Glomus versiforme. When first mentioned in the text, the authority must be added to the scientific name. If possible, a strain number should be provided in Materials and Methods.
Materials and Methods:
It is not clear how the inoculum should produce mycorrhizas given only 30 minutes time of contact. Were the seeds separated from the inoculum after 30 minutes?
Seeding time must be more specific than winter and summer.
Discussion:
217: "Our study showed no colonization by AMF found in the roots of the nonAMF seedlings,"
Table 1 however, does show some colonization, although significantly lower.
The remainder of this paragraph should be edited for clarity.
The whole first paragraph of p12 is a discussion of literature results, with little or no reference to the authors' findings.
10/321: Only one soil sample was collected?
Comments on the Quality of English Language
The manuscript is generally readable, but with some weaknesses in grammar and sentence structure. Some paragraphs are difficult to read and confusing.
Round 2
Reviewer 2 Report
Comments and Suggestions for Authors
This paper does not explain how the spores of AMF are produced. My question is not where AMF came from (who provided it), but how the author, after obtaining the strain, cultured it to produce spores and then took the spores to conduct the experiments mentioned in the draft. Spores are used in this experiment, and it must be noted in detail how the strain was cultured for sporulation.
“Compost tea was prepared by brewing compost and water at a ratio of 1:10 w/v (compost: water) with continuous aeration for 48hour,” The authors did not mention how to prepare the compost. If it is a product everyone can buy, it is necessary to indicate the item number or product number. If not, the production method must be indicated in this paper.
Reviewer 4 Report
Comments and Suggestions for Authors
From the first round of review, not all items were corrected:
Point 2 was only partly fixed. The parts without strikeout were not corrected.
"Point 2: The scientific name is Glomus versiforme. When first mentioned in the text, the authority must be added to the scientific name. If possible, a strain number should be provided in Materials and Methods.
Some paragraphs with poor English language grammar, e.g. the first paragraph of 2.2.2. I'd recommend an English editor to carefully edit the entire manuscript.
Comments on the Quality of English LanguageSome paragraphs with poor English language grammar, e.g. the first paragraph of 2.2.2. I'd recommend an English editor to carefully edit the entire manuscript.
